# BALSA: Benchmarking Active Learning Strategies for Autonomous Laboratories

## Abstract

Accelerating scientific discoveries holds significant potential to address some of the most pressing challenges facing society, from mitigating climate change to combating public health crises, such as the growing antibiotics resistance. The vast and complex nature of design parameter spaces makes identifying promising candidates both time-consuming and resource-intensive, rendering conventional exhaustive searches impractical. However, recent advancements in data-driven methods, particularly within the framework of "active learning," have led to more efficient strategies for scientific discovery. By iteratively identifying and labeling the most informative data points, these methods function in a closed loop, guiding experiments or simulations to accelerate the identification of optimal candidates while reducing the demand for data labeling. Despite these advancements, the lack of standardized benchmarks in this emerging field of autonomous scientific discovery impedes progress and limits its potential translational impact. To address this, we introduce BALSA: a comprehensive benchmark specifically designed for evaluating various search algorithms applied in autonomous laboratories within the active learning framework. BALSA offers a standardized evaluation protocol, provides a metric to characterize high-dimensional objective functions, and includes reference implementations of recent methodologies, with a focus on minimizing the data required to reach optimal results. It provides not only a suite of synthetic functions or controlled simulators but also real-world active learning tasks in biology and materials science — each presenting unique challenges for autonomous laboratory tasks.[1]

## 1 Introduction

Designing proteins or materials with specific properties—ranging from antibiotic resistance to superconductivity—represents a crucial frontier in addressing critical scientific and societal challenges (Hamidieh, 2018; Varmus et al., 2003; Merchant et al., 2023). Traditionally, scientists have approached these design processes by generating hypotheses based on prior knowledge and past data. These hypotheses are then tested using experimental protocols within constrained budgets. However, this approach is often inefficient, time-consuming, and limited by human ingenuity and errors. In recent years, the integration of data-driven methods with automated laboratory setups has accelerated discovery across various fields, ranging from the design of proteins or DNA sequences in biology or the discovery of functional materials (Coley et al., 2019; Rao et al., 2022; Szymanski et al., 2023; Rapp et al., 2024).

One of the most promising innovations in this field is the self-driving laboratories (SLs), which leverage active learning (AL) algorithms to autonomously guide experimentation and accelerate scientific discovery (Häse et al., 2019; Kang et al., 2019; Abolhasani & Kumacheva, 2023). Advances in AL offer the potential to significantly enhance the exploration of larger regions within the expansive search space, thus improving efficiency and effectiveness in experimental designs and optimization processes, as shown in Figure 1a. Given that the underlying model of objective function (or the validation source) is often intractable, and only limited data are available, a typical approach is to develop a surrogate model to approximate the distribution of objective function. This surrogate model

---

[1]Our code can be found at `https://github.com/anonymized`

is then used iteratively to optimize the design, serving as a stand-in for the objective function in the optimization process. The key components of SLs (or AL pipelines) are illustrated in Figure 1b.

Despite significant progress, many strategies to explore the search spaces, including exact and heuristic approaches, often struggle to adapt and scale to high-dimensional and non-linear scenarios found in many science applications (Frazier, 2018). Bayesian Optimization (BO) and its variants (Shahriari et al., 2016; Bubeck et al., 2011; Springenberg et al., 2016), have emerged as popular alternatives that learn a Bayesian model of the objective function and sample the best candidates using an uncertainty-based technique such as Thompson sampling (Shahriari et al.). While these approaches perform well in low-dimensional spaces, their effectiveness diminishes in more complex, higher-dimensional settings (Frazier, 2018). More recently, tree search methods, which are the key component of many revolutionary AI algorithms such as AlphaGo (Silver et al., 2016), have been applied to design problems. These methods iteratively partition the search space (Kim et al., 2020a) and employ local surrogate models to approximate the promising search subspace (Eriksson et al., 2019). However, their success is often contingent on the quality of these local models, and they also struggle with the curse of dimensionality (Wang et al., 2020b).

Moreover, the intricate interplay between surrogate models and search strategies within AL pipelines, coupled with the growing number of scientific applications, has made it increasingly difficult to compare and track progress effectively. Different methods are often proposed and evaluated on distinct tasks with varying evaluation protocols, leading to inconsistent benchmarks. To the best of our knowledge, no unified benchmark or systematic investigation currently exists to evaluate and compare these algorithms across AL strategies. This paper addresses this gap by proposing a standardized benchmark that enables a fair comparison of state-of-the-art AL strategies, ensuring more consistent progress in scientific discovery.

**Our contributions** We conduct a systematic evaluation of the key algorithmic components and their interactions within AL pipelines. Our main contributions are summarized as follows:

- We propose an AL pipeline specifically tailored for real-world self-driving laboratory environments. The pipeline is designed with the following key objectives: (i) to emulate the iterative, step-by-step process characteristic of real-world self-driving tasks; (ii) to leverage surrogate models for the efficient approximation of complex systems in data-scarce scenarios; and (iii) to address the unique challenges associated with low-data regimes, ensuring robust performance under limited data availability.

- We introduce a suite of 6 standardized synthetic tasks and 11 baseline methods to systematically evaluate a broad range of current AL pipelines and the respective surrogate models.

- We design and implement four real-world tasks to evaluate the proposed pipeline: (i) neural network architecture search to optimize model performance; (ii) the lunar landing problem, simulating complex control dynamics; (iii) a biology task utilizing AlphaFold2 as a virtual simulator for protein design, demonstrating applications in computational biology; and (iv) a materials science task focused on resolution optimization of scanning transmission electron microscopes, leveraging professional open-source simulation software for advanced imaging applications.

- Through a systematic investigation and a large-scale empirical study, we introduce a novel metric that quantifies the characteristics of objective landscapes across diverse design problems. This metric offers valuable insights into the behavior and performance of AL methods, enabling a more nuanced understanding of their effectiveness in complex settings.

- Based on these empirical findings, we highlight three critical areas for advancing self-driving labs within the AL pipeline: (i) understanding the interplay between surrogate model and search strategy in relation to the objective landscape, (ii) ensuring reproducibility of algorithmic performance across a wide variety of synthetic and real-world tasks, and (iii) developing methods that handle the optimization problems with limited data availability.

## 2 PROBLEM STATEMENT

The goal of SL tasks is to iteratively identify and label the most informative data points, discovering optimal candidates while minimizing labeling efforts. Figure 1b illustrates the general protocol of

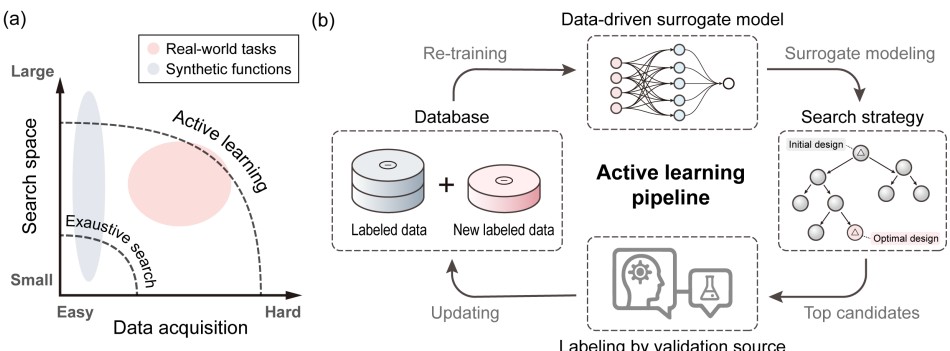

Figure 1: Overview of key components in active learning for self-driving labs. (a) Active learning can address problems with large search space and expensive data acquisition. (b) The goal of the active learning task is to iteratively and autonomously improve solutions. Beyond synthetic functions, the proposed BALSA utilizes i) AlphaFold2 as a simulator for biology applications and ii) open-source scanning transmission electron microscopes (STEM) simulators for materials science applications.

an active learning algorithm, which comprises four components: (i) database, (ii) surrogate model that accurately represents the complex relationships in the data, (iii) search model that utilizes the surrogate model to guide the search for an optimal single state, and (iv) validation source which can provide the ground-truth.

Without loss of generality, assuming that we search for global minima of a function $f$ without explicit formulation and its specific solution $\mathbf{x}^*$ :

$$\mathbf{x}^* = \underset{\mathbf{x} \in X}{\mathrm{argmin}}\, f(\mathbf{x}) \tag{1}$$

where $\mathbf{x}$ is the input vector and $X$ is defined as the search space, typically $\mathbb{R}^n$, and $n$ is the dimension. $f$ is the deterministic function that maps the input $\mathbf{x}$ to the label, which can either be an exact function that provides ground-truth labels or a data-driven surrogate model $\hat{f}$ learned through the dataset $\mathcal{D} = \{(\mathbf{x}_i, y_i)\}_i^N$, in which $N$ is the number of labels and $y_i$ is the label of $\mathbf{x}_i$. It is noteworthy that this function is not limited to single-objective problems, it can be a product of multiple functions as long as it solely depends on $\mathbf{x}$, which makes it a multi-objective task.

## 3 RELATED WORK

**Self-driving labs**   There has been a surge of interest in developing SLs across various applications in all areas of science. Ranging from organic small molecules and compounds (Li et al., 2015; Coley et al., 2019) to synthetic biology (Martin et al., 2023) and drug discovery (Saikin et al., 2019) to chemistry (Jablonka et al., 2024) including multi-step chemistry (Epps et al., 2020; Seifrid et al., 2022; Boiko et al., 2023; Volk et al., 2023), reaction optimization (Torres et al., 2022; Angello et al., 2022), copolymer (Reis et al., 2021) or chemical synthesis (Manzano et al., 2022) synthesis as well as and material science (Szymanski et al., 2023; Merchant et al., 2023) including solid state materials (Szymanski et al., 2023), clean energy (Tabor et al., 2018) or thin films (Ludwig, 2019). Due to the rapid pace of development and interest across various disciplines, we can only include a limited selection.

A curated and up-to-date list across application areas and a broad overview of SLs including applications, software packages, or hardware is provided by the Canadian Acceleration Consortium (Consortium).

**Benchmarks**   Different Benchmarks have been proposed for black-box optimization. Design bench Trabucco et al. (2022) proposed a benchmark for offline model-based optimization. Further benchmarks include robotics systems (Ginsburg et al., 2023) or simple multi-tool motion platforms (jub). Other works developed codebases for optimization algorithms and libraries without downstream tasks or datasets (Rapin & Teytaud, 2018). The traditional optimization benchmark primarily focuses

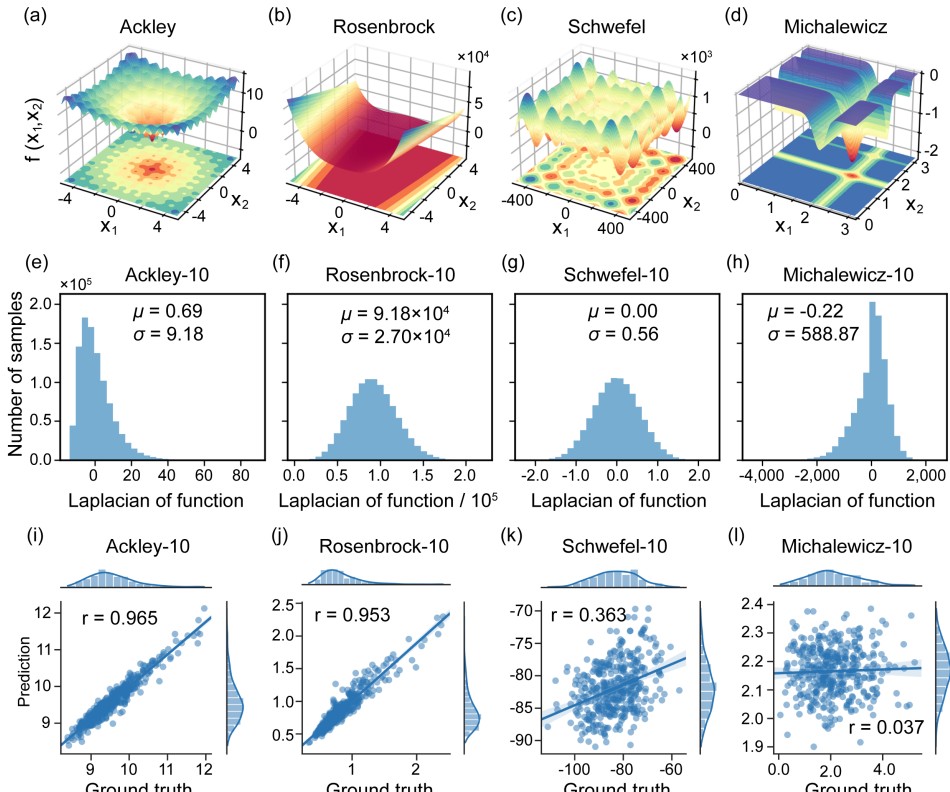

Figure 2: Objective landscapes of different synthetic functions with distinct topological characteristics. Visualization of 2D objective landscapes with (a) Ackley, (b) Rosenbrock, (c) Schwefel, and (d) Michalewicz in their 2D forms. Histograms (frequency distributions) of Laplacian of function $s$ for (e) Ackley, (f) Rosenbrock, (g) Schwefel, and (h) Michalewicz, where each function is in 10-dimension with 1 million samples uniformly sampled from the parameter space. Joint plots of the ground truth function values (x-axis) and the surrogate model predictions (y-axis) for (i) Ackley, (j) Rosenbrock, (k) Schwefel, and (l) Michalewicz, where $r$ denotes the Pearson correlation coefficient. Note that some of the functions are re-scaled to achieve better fitting (see Supplementary S.3 for more details.)

on minimizing the number of function evaluations required to reach the global optimum and the objective often focuses on the optimization of trajectory planning,Our benchmark suite employs the same synthetic function but with a distinct objective. By leveraging synthetic functions with known global optima, our goal is to evaluate the number of data points required by an AL algorithm to converge to these optima. This approach provides an inexpensive means of assessment, offering critical insights into the algorithm's efficiency and effectiveness in optimization tasks across diverse contexts.

## 4 SYNTHETIC BENCHMARKS

Our benchmark suite includes 6 carefully selected functions: Ackley, Rastrigin, Rosenbrock, Griewank, Schwefel, and Michalewicz. The primary objective for these synthetic functions is to identify their global minima with a minimum number of sample acquisitions. Unlike traditional optimization algorithms, which are often parallelizable and primarily focus on minimizing the number of function evaluations required to reach the global optimum, our benchmark study uses these synthetic functions to mimic the complex data distributions generated by various validation sources. The process is iterative, with each iteration allowing only 20 data points to be sampled from the synthetic function tasks. This constraint necessitates the development of an effective learning-based surrogate model. These synthetic functions can serve as valuable test cases for understanding the properties of real-world SL tasks across diverse conditions using different search algorithms with

surrogate models within the AL pipeline. We explore a statistical feature to characterize different objective landscapes that may pose challenges for the AL algorithms. Here, we focus on four key functions: Ackley, Rosenbrock, Schwefel, and Michalewicz, as these functions are characterized by their distinct objective landscapes.

**Landscape characterization** Understanding the topology of an objective function is crucial for evaluating the performance of learning-based surrogate models within the AL pipeline. For instance, a machine learning model often exhibits a less-satisfactory performance on a flat landscape of an objective function, for which most of the values are at the same level, making it difficult for the model to learn and generalize. A poorly performing surrogate model may mislead the search methods, ultimately resulting in sub-optimal outcomes. Figure 2 (a-d) visualizes the objective landscapes of the corresponding synthetic functions in their 2D forms. Ackley shows a rugged but funneled topology, while Rosenbrock exhibits a long valley with numerous local minima. Schwefel presents a complex multi-funnel topology, whereas Michalewicz has sharp drops on a rather flat landscape (The mathematical formula can be found in Supplementary S.1).

However, characterizing high-dimensional objective functions poses additional challenges due to their inherent sparsity and non-convexity. To better understand the relationship between the landscape of the objective function and the performance of the surrogate model, we introduce a landscape flatness. This metric uses random sampling and discrete Laplacian operator to quantify the flatness of the objective landscape. While the metric provides valuable empirical insights, we acknowledge its limitations in theoretical rigor and aim to explore a more comprehensive analysis in future work.

**Laplacian of function** Let $\mathbf{x} = [x_1, ..., x_i, ..., x_n]$ be a $n$-dimensional input of the function. The discrete Laplacian operator at a high-dimensional position $\mathbf{x}$ can be defined as:

$$s_x = \sum_{i=1}^{n} \frac{\partial^2 f}{\partial x_i^2} \approx \sum_{i=1}^{n} \frac{f(x_i + \epsilon) + f(x_i - \epsilon) - 2f(x_i)}{\epsilon^2} \tag{2}$$

where $\epsilon$ is the step size and is set to 0.01 partition of the interval between upper bound and lower bound. The Laplacian of function $s$ is expected to be positive for a locally convex landscape in many of the $i^{th}$ dimensions and to be negative for a locally concave landscape in many of the $i^{th}$ dimensions. A near-zero Laplacian of the function $s$ indicates that the objective function has a rather flat distribution, and there is no gradient on the landscape in many of the $i^{th}$ dimensions.

Figure 2 (e-h) demonstrate the frequency distributions of $s$ and the corresponding mean $\mu$ and standard deviation $\sigma$, where we uniformly sampled 1 million inputs from the individual parameter spaces (in 10D) of the functions. Ackley shows a positively skewed distribution with $\mu$ close to 0 and $\sigma$ of 9.18, suggesting a moderate fluctuation in concavity across all dimensions with some more convex areas (Figure 2e). Rosenbrock shows both large $\mu$ of $9.18 \times 10^4$ and $\sigma$ of $2.70 \times 10^4$, indicating a landscape that is heavily convex anywhere in the landscape domain, with highly anisotropic concavity across all dimensions (Figure 2f). In contrast, Schwefel shows near-zero values for both $\mu$ and $\sigma$, implying a landscape that is generally flat with a rather small, isotropic concavity across all dimensions (Figure 2g). Interestingly, Michalewicz shows a $\mu$ close to 0 and an abnormally large $\sigma$, implying that the landscape is flat with some small areas being dramatically concave or convex (Figure 2h).

**Landscape flatness** To quantitatively measure the flatness of the landscape, we introduce a metric landscape flatness $\omega$ based on the mean $\mu$ and variance $\sigma$ of the frequency distributions of $s$, which is defined as:

$$\omega = \sqrt{\frac{\sigma}{|\mu|}}. \tag{3}$$

Ackely-10 and Rosenbrock-10 have $\omega$ of 3.62 and 0.54, respectively, whereas $\omega$ of Schwefel-10 and Michalewicz-10 are 37.07 and 54.47, respectively, indicating that the overall landscape is rather flat. Indeed, Figure 3 suggests that the flatness $\omega$ is highly correlated to the performance of the surrogate model, the functions with lower $\omega$ are easier to be learned than those with higher $\omega$.

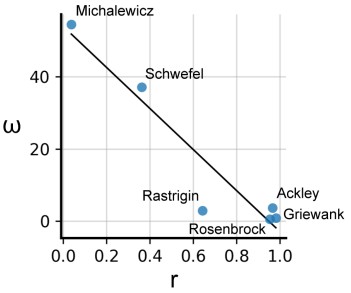

Figure 3: Correlation between Pearson correlation coefficient ($r$) and flatness ($\omega$).

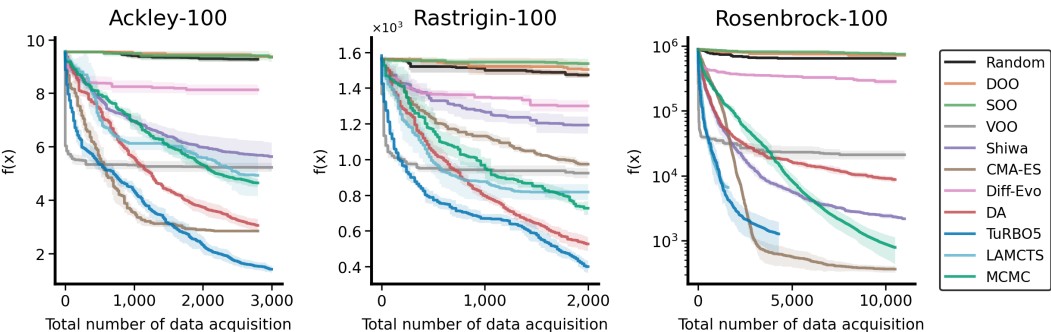

Figure 4: Evaluation of sampling efficiency for Ackley, Rastrigin, Rosenbrock in 100-dimension. No single method demonstrates consistent superiority across all scenarios.

**Surrogate model training**   A key challenge for AL methods with surrogate models is to learn a good approximator of the objective function with only a few samples. Figure 2 (i-l) presents the correlations of the ground truth function values and the surrogate model (i.e. neural network in this case) predictions for different functions (all in their 10D forms). Each surrogate model $\hat{f}$ was trained on a dataset $\mathcal{D} = \{(\mathbf{x}_i, f(\mathbf{x}_i)\}$ of inputs $\mathbf{x}_i$ and the corresponding function value $f(\mathbf{x}_i)$ (see Supplementary S.3 for more details.) It can be observed that surrogate models generalize better on landscapes with gradients (i.e., Ackley and Rosenbrock), and worse on flat landscapes (i.e., Schwefel and Michalewicz.). It is likely that a surrogate model requires many more samples to generalize in the low $\omega$ scenario.

**Data sampling efficiency**   Figure 4 shows the history of the AL performance to evaluate the sampling efficiency of the algorithms. Here, 11 methods are benchmarked against the current minimum across different data acquisition scenarios. The results reveal that no single method consistently outperforms others across all situations. Notably, TuRBO5 achieves the best performance on the Ackley-100 and Rastrigin-100 tasks, while CMA-ES excels in the Rosenbrock-100 task.

## 5   REAL-WORLD BENCHMARK TASKS

Many real-world tasks can be treated as VLs, where high-fidelity simulators are combined with learning models, automatically optimizing designs to achieve better mechanical, physical, or chemical properties within a virtual environment. VLs are essential across a multitude of complex real-world systems, particularly when experiments are associated with prohibitive costs and extensive design spaces. The virtual tasks included in VLs can be framed as typical AL problems. In this work, we focus on four benchmark tasks within SLs: neural network architecture search, lunar landing problem, cyclic peptide design and optimization of electron ptychography reconstruction. These benchmark tasks are selected because (i) they are supported by accurate high-fidelity simulators, (ii) they address optimization problems with single or multiple objectives in the fields of materials science and biology, and (iii) they can be executed within reasonable time and computational resources.

**Neural network architecture search**   NAS is an automated approach for identifying optimal neural network architectures by systematically exploring and evaluating a wide range of network configurations to achieve superior performance on a specific task. Detailed experimental setups and methodologies are provided in Supplementary S.5.

**Lunar landing problem**   The Lunar Lander problem is a widely recognized benchmark environment in the OpenAI Gym toolkit, frequently utilized in reinforcement learning research to evaluate control strategies. The task involves controlling a simulated lunar module to achieve a safe landing on the moon's surface. The environment provides four discrete action options: (i) do nothing, (ii) fire the left engine, (iii) fire the main engine, and (iv) fire the right engine. While this problem is traditionally framed as a trajectory planning task with cumulative objectives, we reformulate it into a non-cumulative optimization problem by fixing the initial conditions. The goal is to design an

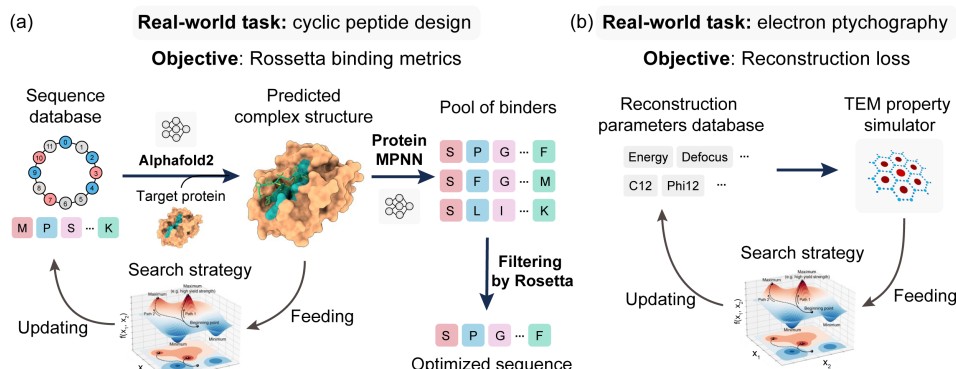

Figure 5: Pipelines of two chosen real-world tasks: (a) cyclic peptide design and (b) electron ptychography.

optimal sequence of 100 discrete actions to maximize the reward, where the action space includes 0 (do nothing), 1 (fire left engine), 2 (fire main engine), and 3 (fire right engine). To ensure consistency, the environment reset seed is fixed at 42 to generate a consistent initial state. Further details on the experimental setup can be found in Supplementary S.6.

## 5.1 CYCLIC PEPTIDE DESIGN

**Background**  Cyclic peptides are a class of compounds that have garnered significant attention as therapeutic agents due to their enhanced stability, high specificity, and excellent membrane permeability. These properties make them particularly effective in targeting traditionally "undruggable" protein surfaces (Vinogradov et al., 2019). The amino acids (AAs) in cyclic peptides are interconnected by amides or other chemically stable bonds, which can be chosen from the 20 standard AAs or various non-standard ones, creating a high-dimensional and complex sequence design space (Zorzi et al., 2017). Here, the task is more specific than general protein design: it involves designing a specialized type of protein with therapeutic applications. This protein is required to exhibit stronger interactions with its target, such as higher binding affinity. Such a task can be framed as an optimization problem. However, even for a relatively simple 16-residue sequence, the combinatorial search space includes $16^{20}$ possible configurations. The intricate and nonlinear relationship between protein sequence and functional properties further complicates the challenge, making it a suitable benchmark for testing advanced methodologies. An additional advantage of this setup is the availability of natural binders as a reference for comparison. Traditionally, one often needs to conduct high-throughput wet lab experiments, synthesizing thousands of cyclic peptides before discovering one that can specifically bind to a desired protein (Gang et al., 2018). VL can accelerate this discovery process by narrowing the potential candidates to a few dozen, drastically reducing the cost.

**Optimization target**  The optimization target of cyclic peptide design is defined as follows:

$$Target = SC \cdot dSASA \tag{4}$$

The SC value ranges from 0 to 1, referring to how well the surfaces of two proteins fit geometrically together at their interface; $dSASA$ measures the size of the interface (in units of $\mathring{A}^2$). a larger $dSASA$ reflects a more extensive interface area. Further details regarding the dataset and simulation settings can be found in Supplementary S.7.

## 5.2 ELECTRON PTYCHOGRAPHY

**Background**  Electron ptychography is a phase-contrast imaging technique capable of resolving nanostructures at a sub-angstrom resolution. Electron ptychography is widely used for specimens thicker than a monolayer (Cowley & Moodie, 1957) and sensitive materials vulnerable to beam-induced damage (Song et al., 2019). However, electron ptychography relies on a careful selection of various reconstruction parameters, such as physical, optimization, and experimental parameters, which affect the quality and accuracy of the retrieved transmission function. The parameter space

Table 1: Evaluations of AL methods on synthetic functions with the usage of surrogate model, where the values with bold texts denote the best optimization result across all the methods. Results are averaged over 5 trials, and ± denotes the standard deviation.

| | Ackley-20 | Ackley-100 | Rastrigin-20 ($\times 10^2$) | Rastrigin-100 ($\times 10^3$) | Rosenbrock-20 ($\times 10^4$) | Rosenbrock-100 ($\times 10^4$) | Schwefel-20 ($\times 10^3$) | Michalewicz-20 |
|---|---|---|---|---|---|---|---|---|
| $f(\mathbf{x}^*)$ | 0.00 | 0.00 | 0.00 | 0.00 | 0.00 | 0.00 | 0.00 | -19.63 |
| **Random** | 7.59 ± 0.17 | 9.23 ± 0.13 | 2.18 ± 0.15 | 1.47 ± 0.016 | 2.380 ± 0.119 | 64.60 ± 0.936 | 5.50 ± 0.11 | -6.11 ± 0.42 |
| **TuRBo5** | 0.37 ± 0.14 | **1.73 ± 0.18** | **0.52 ± 0.04** | **0.40 ± 0.034** | **0.003 ± 0.000** | 0.127 ± 0.066 | 2.84 ± 0.79 | **-11.34 ± 1.20** |
| **LaMCTS** | 1.96 ± 0.75 | 5.05 ± 0.73 | 0.80 ± 0.30 | 0.82 ± 0.044 | 0.008 ± 0.005 | 0.652 ± 0.098 | 3.32 ± 0.33 | -7.66 ± 0.44 |
| **CMA-ES** | 0.75 ± 0.09 | 2.85 ± 0.04 | 0.78 ± 0.03 | 0.97 ± 0.017 | 0.006 ± 0.004 | **0.037 ± 0.004** | 5.28 ± 0.44 | -6.38 ± 0.33 |
| **Diff-Evo** | 6.43 ± 0.16 | 8.13 ± 0.19 | 1.88 ± 0.12 | 1.30 ± 0.032 | 0.797 ± 0.115 | 28.30 ± 2.690 | 5.10 ± 0.17 | -6.05 ± 0.73 |
| **DA** | **0.00 ± 0.00** | 3.28 ± 0.19 | 1.29 ± 0.06 | 0.53 ± 0.039 | 0.005 ± 0.003 | 0.908 ± 0.088 | 2.38 ± 0.39 | -10.03 ± 0.77 |
| **Shiwa** | 4.43 ± 0.07 | 5.78 ± 0.52 | 2.48 ± 0.02 | 1.19 ± 0.047 | 2.266 ± 0.146 | 0.240 ± 0.022 | 5.49 ± 0.32 | -6.65 ± 1.13 |
| **MCMC** | **0.00 ± 0.00** | 4.79 ± 0.16 | 0.89 ± 0.27 | 0.73 ± 0.038 | 0.011 ± 0.006 | 0.088 ± 0.036 | **2.11 ± 0.86** | -9.74 ± 1.18 |
| **DOO** | 7.17 ± 0.37 | 9.44 ± 0.19 | 2.22 ± 0.14 | 1.50 ± 0.044 | 1.640 ± 0.456 | 72.22 ± 2.700 | 5.56 ± 0.29 | -6.13 ± 0.28 |
| **SOO** | 7.75 ± 0.18 | 9.40 ± 0.17 | 2.24 ± 0.08 | 1.54 ± 0.027 | 2.760 ± 0.744 | 76.30 ± 2.700 | 2.89 ± 2.18 | -6.34 ± 1.17 |
| **VOO** | 2.44 ± 0.49 | 5.23 ± 0.17 | 1.03 ± 0.13 | 0.92 ± 0.028 | 0.006 ± 0.000 | 2.107 ± 0.324 | 5.38 ± 0.08 | -7.98 ± 0.79 |

All benchmark tasks here involve minimization objectives.

The asterisk (*) represents the global minimum of the function.

is vast and complex, and the optimal choice depends on the specific configuration of dataset and measurement conditions. Although some algorithms have been applied to this task (such as Bayesian optimization using Gaussian process (Cao et al., 2022)), the parameter selection process still strongly relies on expert knowledge and trial-and-error, which limits the efficiency and applicability of electron ptychography.

**Optimization target** The goal of this task is to optimize the reconstruction parameters within the electron ptychography algorithm to retrieve the best quality of phase of the transmission function within the atomic lattice. This requires solving a non-convex problem in a 15D parameter space in our case (see Supplementary S.8 for details). Specifically, the objective function is the normalized mean square error (NMSE) between the positive square-root of the measured diffraction pattern $I_M$ and the modulus of the Fourier-transformed simulated exit-wave $\Psi$, which can be formulated as:

$$\frac{1}{N} \sum_{i}^{N} \left| \sqrt{I_{M(i)}(\mathbf{u})} - |\mathcal{F}[\Psi_i(\mathbf{r})]| \right|^2 \tag{5}$$

where $\mathbf{r}$ and $\mathbf{u}$ denote the real- and reciprocal-space coordinate vectors, respectively, $N$ is the total number of the measured diffraction patterns, and the operator $\mathcal{F}$ represents a Fourier transform. Further details regarding the dataset, simulation settings and evaluation metrics can be found in Supplementary S.8.

# 6 BENCHMARK RESULTS

## 6.1 SYNTHETIC FUNCTION TASKS

We benchmark 11 state-of-the-art search methods (including Random Search) alongside neural network as the surrogate model on synthetic function tasks within the AL pipeline. These methods span a wide range of algorithm categories, including Dual Annealing (DA (Pincus, 1970)), Evolutionary Algorithm (CMA-ES (Hansen et al., 2003), Differential Evolution (Diff-Evo (Storn & Price, 1997)), Shiwa (Liu et al., 2020)), Bayesian Optimization (BO (Gardner et al., 2014), TuRBO (Eriksson et al., 2020)), Monte Carlo Tree Search (LaMCTS(Wang et al., 2020a), DOO (Munos, 2011), SOO (Munos, 2011), and VOO (Kim et al., 2020b)). The implementation settings of each AL algorithm can be found in Supplementary S.4. Our evaluation covers all functions in their 20D forms, as well as the Ackley, Rastrigin, and Rosenbrock functions in both 20D and 100D forms. The results led to two key insights. First, these methods are more effective with lower-dimensional functions, but their performance diminishes as dimensionality increases. Second, search methods tend to work better on functions that have well-fitting surrogate models (i.e., Ackley and Rosenbrock), while they perform less well or even not better than random sampling with poorer surrogate model fittings (i.e. Schwefel and Michalewicz, as shown in Figure 2). The observed variance primarily arises from data sparsity

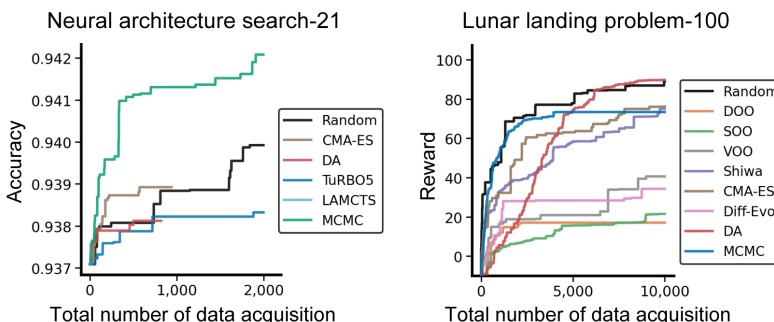

Figure 6: Benchmarks of sampling efficiency for Neural Architecture Search (NAS) in 21-dimension and lunar landing problem in 100-dimension. Note that both problems involve maximization objectives.

Table 2: Evaluations on two real-world tasks. Shape complementarity (SC) and the change in Solvent Accessible Surface Area ($dSASA$) are used for cyclic peptite design, and normalized mean square error, object reconstriction error and probe reconstriction error are used for ptychographic reconstruction on the MoS2 dataset. Uppward arrow (↑) and downward arrow (↓) indicate maximization and minimization tasks, respectively. Results are averaged over 3 trials, and ± denotes the standard deviation.

| | Cyclic peptide design | | | | | | Electron ptychography | | |
|---|---|---|---|---|---|---|---|---|---|
| | 4kel-SC ↑ | 4kel-dSASA ↑ | 4kel-Target ↑ | 7j2k-SC ↑ | 7j2k-dSASA ↑ | 7j2k-Target ↑ | NMSE ↓ | Object recon. error ↓ | Probe recon. error ↓ ($\times 10^{-3}$) |
| Reference* | 0.77 | 1505 | 1156 | 0.67 | 865 | 582 | 0.079 | 0.048 | 0.35 |
| Diff-Evo | 0.72 ± 0.05 | 1464 ± 65 | 1046 ± 69 | 0.66 ± 0.04 | 923 ± 72 | 613 ± 61 | 0.283 ± 0.005 | 0.102 ± 0.008 | 2.96 ± 0.34 |
| DA | 0.70 ± 0.03 | 1556 ± 32 | 1096 ± 48 | 0.65 ± 0.04 | 894 ± 59 | 570 ± 19 | 0.313 ± 0.005 | 0.118 ± 0.011 | 3.05 ± 0.27 |
| TuRBO | 0.71 ± 0.03 | 1501 ± 37 | 1059 ± 55 | 0.63 ± 0.01 | 904 ± 42 | 572 ± 17 | 0.275 ± 0.000 | 0.104 ± 0.001 | 2.60 ± 0.08 |
| BO | 0.72 ± 0.02 | 1431 ± 14 | 1035 ± 22 | 0.60 ± 0.03 | 908 ± 56 | 546 ± 57 | 0.300 ± 0.000 | 0.097 ± 0.000 | 3.28 ± 0.00 |

*Reference denotes "native" for cyclic peptide design and "expert reconstruction result" for electron ptychography.

associated with high dimensionality. Within our active learning pipeline, we train a surrogate model that serves as the basis for exploration and optimization by search algorithms. Notably, the search algorithm operates without direct access to ground truth labels, making the random initialization of the surrogate model's training dataset a critical factor influencing the outcomes. Variations in these initializations yield distinct surrogate models, which in turn contribute to increased variance across trials. This effect is particularly pronounced in high-dimensional problems, where greater variance is anticipated due to the exacerbated sparsity.

## 6.2 REAL-WORLD TASKS

For the NAS and lunar landing problem, we benchmark the results using six to nine different AL methods. For biology and materials science tasks, we evaluate the performance of four selected AL methods: Diff-Evo, DA, TuRBO5, and BO. Each task is subjected to three independent trials to ensure robust results, with each AL method having a fixed number of oracle function evaluations.

**Neural Architecture Search and Lunar Landing Problem**  Figure 6 shows benchmark results of both real-world problems. As for NAS, We benchmark the problem with six optimization algorithms: Random Search, MCMC, CMA-ES, DA, LAMCTS, and TuRBO5, where MCMC dominates and rapidly reaches 0.941 with 500 data acquisitions. Regarding the lunar landing, we evaluate this problem using nine algorithms: Random Search, DOO, SOO, VOO, Shiwa, CMA-ES, Diff-Evo, DA, and MCMC.

**Cyclic peptide design**  Table 2 presents the results of the AL methods for different metrics. In the cyclic peptide design task, global optima is unknown, and therefore any method that yields the target value exceeding the native complex (denoted as "Reference" in Table 2) can be considered a 'success'. According to this criterion, none of the tested AL methods succeeded in finding a better binder for protein (pdbid: 4kel), and only Diff-Evo achieved a better design for protein 7j2k. However, it is

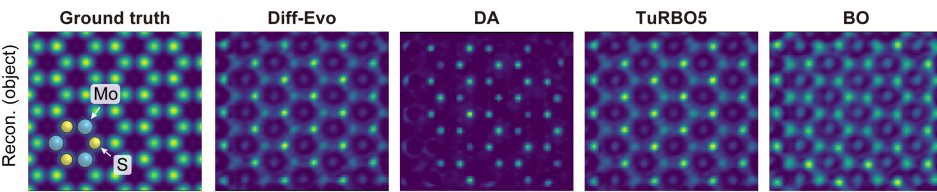

Figure 7: Benchmarking the electron ptychography task: visualization of the reconstructed phases (of the object transmission functions) with parameters obtained from the corresponding AL methods. No single method achieves results comparable to the ground truth.

noteworthy that in this type of design task, native does not represent the best designs. Figure S2 illustrates the complex with the highest target value optimized by the AL method for protein 4kel. All these complexes contain hydrophobic residues that fit into the protein pocket, contributing to the high target values. More detailed settings about AL methods can be found in Supplementary S.7

**Electron ptychography**    Table 2 summarizes the performance of AL methods on ptychographic reconstructions of the $MoS_2$ dataset, where "Reference" denotes the expert reconstruction results for a single-layer $MoS_2$ dataset with the same aberration settings. It is observed that all AL methods do not achieve the optimal reconstruction of both object and probe functions. However, TuRBO5 and Diff-Evo can attain lower NMSE values and have generally more physically sensible reconstructions for the phases of the object transmission functions. As shown in Figure 7, despite not being perfect, both AL methods (Diff-Evo and TuRBO5) can resolve the atomic contrasts of heavy Molybdenum (brighter) atoms and light Sulfur atoms (darker). On the other hand, DA and BO present higher NMSE values and are considered worse in ptychographic reconstruction. We note that although not able to fully resolve atomic contrasts from different atoms, BO has the lowest object reconstruction error and can retrieve an object transmission function with general atomic signals. More detailed analyses are included in Supplementary S.8.

## 7    DISCUSSION

Biology and material science applications represent exciting application areas with tremendous potential for the development of self-driving labs. However, the absence of standardized benchmarks and evaluation protocols has hindered the accurate tracking of progress. To address this, we design an active learning pipeline that tailors to self-driving lab settings, including (i) iterative process, (ii) use of surrogate models and(iii) low-data regime. Our benchmark BALSA is a comprehensive resource that includes (i) a codebase, (ii) a suit of synthetic tasks, and (iii) two complex tasks with controlled simulators and two real-world applications in biology and materials science. It features a large-scale empirical evaluation, providing a template for reproducible research and for systematically advancing the performance of algorithms across disciplines. Virtual labs and high-fidelity simulators have the potential to reduce the need for costly and time-consuming real-world experiments. Our extensive evaluation highlights current limitations and indicates promising directions for future research, including developing methods for hyperparameter selection with network-based surrogate models and scaling approaches to very high dimensions.

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

## S  SUPPLEMENTARY MATERIALS

### S.1  SYNTHETIC FUNCTIONS

The synthetic functions are designed to evaluate and analyze computational optimization approaches. In total, six of them are selected based on their physical properties and topologies. The Ackley function can be written as:

$$f(x) = -a \cdot exp(-b\sqrt{\frac{1}{d}\sum_{i=1}^{d} x_i^2} - exp(\frac{1}{d}\sum_{i=1}^{d} cos(cx_i)) + a + exp(1), \tag{6}$$

where $a = 20$, $b = 0.2$, $c = 2\pi$, and $d$ is the dimension.

The Rosenbrock function can be written as:

$$f(x) = \sum_{i=1}^{d-1}[100(x_{i+1} - x_i^2)^2 + (x_i - 1)^2]. \tag{7}$$

The Rastrigin function can be written as:

$$f(x) = 10d + \sum_{i=1}^{d-1}[x_i^2 - 10\cos(2\pi x_i)]. \tag{8}$$

The three functions are evaluated on the hypercube $x_i \in [-5, 5]$, for all $i = 1, \ldots, d$ with a discrete search space of a step size of 0.1.

The Schwefel function can be written as:

$$f(x) = 418.9828d - \sum_{i=1}^{d} x_i \sin(\sqrt{|x_i|}), \tag{9}$$

where $d$ is the dimension. The function is evaluated on the hypercube $x_i \in [-500, 500]$, for all $i = 1, \ldots, d$ with a discrete search space of a step size of 1.

The Griewank function can be written as:

$$f(x) = \sum_{i=1}^{d} \frac{x_i^2}{4000} - \prod_{i=1}^{d} \cos(\frac{x_i}{\sqrt{i}}) + 1, \tag{10}$$

where $d$ is the dimension. The function is evaluated on the hypercube $x_i \in [-600, 600]$, for all $i = 1, \ldots, d$ with a discrete search space of a step size of 1.

The Michalewicz function can be written as:

$$f(x) = -\sum_{i=1}^{d} \sin(x_i) \sin^{2m}(\frac{ix_i^2}{\pi}), \tag{11}$$

where $d$ is the dimension. The function is evaluated on the hypercube $x_i \in [0, \pi]$, for all $i = 1, \ldots, d$ with a discrete search space of a step size of $10^{-4}$.

### S.2  DATA SAMPLE EFFICIENCY

Figure S1 shows the history of the active learning performance to evaluate the sampling efficiency of the algorithms with 20-dimension. Similar to Figure 4, 11 methods are evaluated against the current minimum across different data acquisition scenarios. Consistent with obversation in high dimensional problems, no single method demonstrates consistent dominance across all tasks. For the Ackley-20 function, DA and MCMC demonstrate rapid convergence to the global minimum of $f(x)$. In the Rastrigin-20 function, TuRBO5 and DA outperforms other approaches. Interestingly, search methods such as TuRBO5 constantly achieves lower values, whereas others, e.g. Diff-Evo, appear to become trapped in local minima.

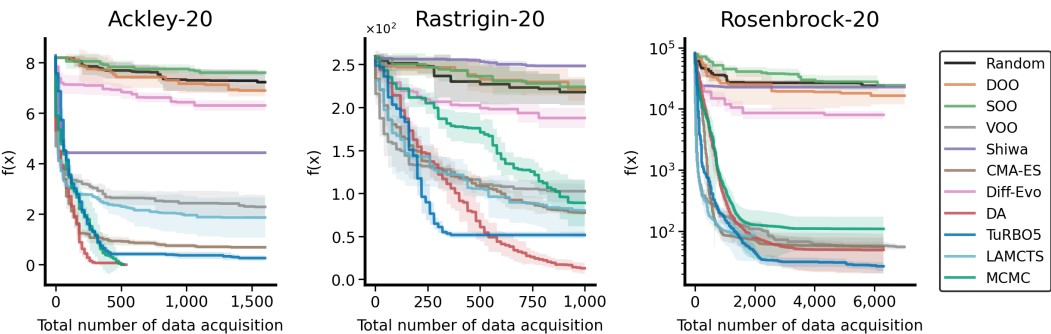

Figure S1: Evaluation of sampling efficiency for Ackley, Rastrigin, Rosenbrock in 20-dimension. No single method demonstrates consistent superiority across all scenarios.

### S.3 SURROGATE MODEL SETUPS

**Training details**   We used 1D convolutional neural networks (1D-CNN) as the surrogate model to fit the synthetic functions. We initiated each surrogate model training with 2,000 uniformly sampled data points from the parameter space of the corresponding synthetic function to train the surrogate model, where 1,600 samples were used for the training set and 400 samples for the testing set. Adam Optimizer was employed with a learning rate of 0.001, and the activation function utilized is the Exponential Linear Unit (ELU). The loss function is the mean square error (MSE) for all synthetic functions except Rastrigin where we used mean absolute percentage error (MAPE). The 1D-CNN model is trained for 500 epochs with early stopping patience of 30 and a batch size of 64. Additionally, the outputs for some of the functions are transformed to avoid the scaling problem for surrogate model training, the corresponding transformation (if applied) is defined in the corresponding sections as follows.

**Ackley**   The 1D-CNN comprises 5 convolutional layers with filter sizes of 128, 64, 32, 16, and 8 respectively, each using a kernel size of 3. It also includes 2 max-pooling layers with a pooling size of 2, 2 dropout layers with a dropout rate of 0.2, followed by a flatten layer, 2 fully connected layers with 128 and 64 units respectively, and an output layer. To obtain a better fitting, we employed a transformation of $100/(f(\mathbf{x}) + 0.01)$ to the output of the Ackley function $f(\mathbf{x})$.

**Rastrigin**   The 1D-CNN consists of 6 convolutional layers with filter sizes of 256, 128, 64, 32, 16, and 8 respectively. The kernel sizes are 5, 5, 3, 3, 3, and 3 respectively, with strides of 1, 2, 2, 1, 1, and 1 respectively. Following these convolutional layers is a flatten layer, 2 fully connected layers with 128 and 64 units respectively, and an output layer.

**Rosenbrock**   The 1D-CNN comprises 6 convolutional layers with filter sizes of 128, 64, 32, 16, 8, and 4 respectively, each using a kernel size of 3. Additionally, there are 3 max-pooling layers with a pooling size of 2, 2 dropout layers with a dropout rate of 0.2, followed by a flatten layer, 1 fully connected layer with 64 units, and an output layer. To obtain a better fitting, we employed a transformation of $100/(f(\mathbf{x})/100d + 0.01)$ to the output of the Rosenbrock function $f(\mathbf{x})$ in its $d$-dimensional form.

**Griewank**   The model architecture is the same as Rosenbrock. We employed the transformation $10/(f(\mathbf{x})/d + 0.001)$ to the output of the Griewank function $f(\mathbf{x})$ in its $d$-dimensional form.

**Schwefel**   The 1D-CNN consists of 7 convolutional layers with filter sizes of 256, 128, 64, 32, 16, 8, and 4 respectively. The kernel size is set to 5 with a stride of 1 for all layers. These are followed by a flatten layer, 6 fully connected layers with 128, 64, 32, 16, and 8, respectively, and an output layer. We re-scaled the output of the Schwefel function $f(\mathbf{x})$ with a factor of 0.01.

**Michalewicz**   The 1D-CNN comprises 5 convolutional layers with filter sizes of 128, 64, 32, 16, and 8 respectively, each using a kernel size of 3 with a stride of 1. Additionally, there are 3 max-pooling

layers with a pooling size of 2, 2 dropout layers with a dropout rate of 0.2, followed by a flatten layer, 1 fully connected layer with 64 units, and an output layer.

## S.4 AL ALGORITHMS SETUPS

For the benchmark of synthetic function tasks, the AL algorithms were conducted without information on the ground truth oracle functions. The implementations of VOO, SOO, and DOO were sourced from an established repository [1], while the methods including CMA-ES, Differential Evolution (Diff-Evo), and Dual Annealing (DA) were derived from the Scipy optimize module, and Shiwa was obtained from Nevergrad [2]. The implementation of Bayesian Optimization is from [3]. The implementation of TuRBO5 is from [4]. The implementation of LAMCTS is from [5]. All algorithms were employed with the default setting in the reference implementation.

## S.5 ADDITIONAL NEURAL NETWORK ARCHITECTURE SEARCH DETAILS

**Dataset and optimization target**    To benchmark the efficacy of AL algorithms in optimizing neural network structures within the context of active learning, we choose the NAS-Bench-101 dataset (Ying et al., 2019), which contains over 400,000 unique convolutional neural networks along with their corresponding performance metrics, trained on the CIFAR-10 dataset (Hinton et al., 2012). Each neural network is represented by a 7×7 upper-triangular adjacency matrix with up to 9 edges, where nodes represent specific operations and edges denote the connection relationships between these operations. The first operation represents the input, and the last represents the output, while the remaining five components can be selected from 3×3 convolution, 1×1 convolution, or 3×3 max-pooling. The objective of the NAS task is to identify an optimized neural network structure that achieves the highest classification accuracy on the test set (test acc).

**Neural network architecture encoding**    We adopt a truncated 40-bit path-based encoding scheme (White et al., 2021) to represent the neural network structure, where each bit corresponds to a specific path from the input layer to the output layer, incorporating various operators along the way. For optimization algorithms like CMA-ES, Dual Annealing, LAMCTS, and TuRBO5, which require a well-defined search domain, we parameterize the neural network structure into a 36-dimensional vector within the continuous [0, 1] space, as adopted from prior work (Letham et al., 2020). The first 21 entries correspond to the adjacency matrix, where the largest values set the respective elements in the matrix to 1. The remaining 15 entries represent the one-hot encoding of 5 components, each with three possible operations. For MCMC and Random Search, optimization is performed directly at the adjacency matrix level.

**Surrogate model**    We train a 1D-CNN model to map the path encoding into the test acc. The 1D-CNN consists of 5 convolutional layers with filter sizes of 128, 64, 32, 16, and 8, respectively, each using a kernel size of 3. It also includes 2 max-pooling layers with a pooling size of 2, 2 dropout layers with a dropout rate of 0.2, followed by a flatten layer, 2 fully connected layers with 128 and 64 units, respectively, and a final output layer. The loss function used is mean square error (MSE).

**AL algorithm srtups**    The optimization process begins by generating 200 random initial data points from NAS-Bench-101, which are used to train the initial surrogate model. In the active learning loop, optimization algorithms then sample 20 optimized successors by refining the surrogate model, expanding the dataset. The updated surrogate model is subsequently used in the next iteration of the loop, continually improving the optimization process.

- **MCMC**: The acceptance rate is defined as $exp(-\delta/T)$, where $\delta$ represents the difference between the proposal point and the current best point. If $\delta > 0$, indicating the proposal point is better than the current best, the proposal is accepted outright; otherwise, it is accepted

---

[1] https://github.com/beomjoonkim/voot
[2] https://github.com/facebookresearch/nevergrad
[3] https://github.com/bayesian-optimization/BayesianOptimization
[4] https://github.com/uber-research/TuRBO
[5] https://github.com/facebookresearch/LaMCTS

with the calculated acceptance rate. The temperature parameter, T, decreases exponentially with each iteration, starting at an initial value of 0.01, with a half-life of 200 iterations.

- **CMA-ES**: 0.25 $sigma_0$, 300 maxfevals, with other parameters using default settings.

- **DA**: 5 maxiter, 300 maxfun, with other parameters using default settings.

- **LAMCTS**: 40 ninits, 0.1 Cp, 100 iterations, with other parameters using default settings.

- **TuRBO5**: 50 n_init, 300 max_evals, 5 n_trust_regionsm, 10 batch_size, 2000 max_cholesky_size, 50 n_training_steps, with other parameters using default settings.

## S.6 ADDITIONAL NEURAL NETWORK ARCHITECTURE SEARCH DETAILS

**Search algorithms**   The setups of the search algorithms in the AL pipelines are as follows:

- **Random**: Random seed is set to 42.

- **DOO**: 0.1 explr_p with other parameters using default settings.

- **SOO**: Default settings.

- **VOO**: 1 explr_p with other parameters using default settings.

- **Shiwa**: Default settings.

- **CMA-ES**: Default settings.

- **DA**: Default settings.

- **MCMC**: Default settings.

## S.7 ADDITIONAL CYCLIC PEPTIDE DESIGN DETAILS

**Dataset**   Two protein and canonical cyclic peptide complexes, PDBID: 4kel and PDBID: 7k2j, are sourced from the Protein Data Bank (PDB). The former is a 14-amino acid serine protease inhibitor targeting human kallikrein-related peptidase 4 (KLK4) (Riley et al., 2019), while the latter is a cyclic 7-mer peptide interacting with Kelch-like ECH-Associated Protein-1 (KEAP1) (Ortet et al., 2021). For simplicity, we only consider standard amino acids. Therefore, each cyclic peptide is represented as a sequence of integers ranging from 0 to 19, with each number corresponding to a distinct type of standard amino acid, making this a discrete optimization task.

**Pipeline**   The pipeline for cyclic peptide VL consists of three components: (1) AlphaFold2 with cyclic offsets to predict the structure of protein-cyclic peptide complexes (Kosugi & Ohue, 2023); (2) ProteinMPNN to ensure the diversity of designed cyclic peptide sequences (Dauparas et al., 2022); and (3) Rosetta's interface analyzer to evaluate the quality of the designed interface (Leaver-Fay et al., 2011). Given the structure of the desired protein and the corresponding interaction hotspot, the pipeline begins with an optimization method that iteratively searches for the sequence yielding the highest AlphaFold2 pLDDT (predicted Local Distance Difference Test) score, which indicates the confidence level of the predicted structure. The optimized sequence is fed into ProteinMPNN to generate a pool of diverse sequences. Finally, the product of two Rosetta binding metrics—shape complementarity (SC) and the change in Solvent Accessible Surface Area ($dSASA$)—is used to filter the output sequences, with the best-fit design likely to have high SC and $dSASA$ values (Muratspahić et al., 2023).

**Simulation Settings**   The structure of the protein and cyclic peptide complex is predicted using AlphaFold2-multimer with cyclic offsets, as implemented in ColabDesign (Kosugi & Ohue, 2023). ProteinMPNN is also employed in ColabDesign with a batch size of 128. The $SC$ and $dSASA$ values for the predicted structure of the protein and cyclic peptide complex are computed using the PyRosetta Interface Analyzer (Chaudhury et al., 2010).

**AL algorithm setups**   To ensure a fair comparison across AL methods, we limited the number of oracle function evaluations to approximately 1000. The specific settings are detailed as follows.

- **Diff-Evo**: a population size of 15 with a maximum of 1000 function evaluations.

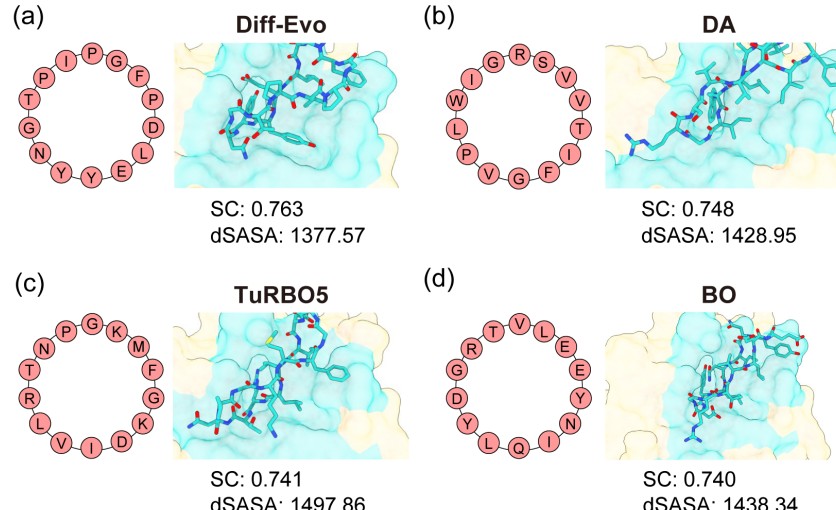

Figure S2: Benchmarking the cyclic peptide design task: visualization of protein 4kel yielded complex results, with the highest target value observed across three trials, where SC and dSASA denotes shape complementarity and change in Solvent Accessible Surface Area, respectively. The left inset illustrates the cyclic peptide sequence, while the right inset presents the interaction map for each method: (a) Diff-Evo, (b) DA, (c) TuRBO5, and (d) BO.

- **DA**: 50 iterations with a maximum of 1000 function evaluations.
- **TuRBO5**: 20 initial samples with 5 trust regions, followed by up to a maximum of 1000 evaluations in batches of 5.
- **BO**: 50 initial samples followed by 950 iterations.

## S.8 ADDITIONAL ELECTRON PTYCHOGRAPHY DETAILS

**Dataset** The dataset is a 4D datacube, comprising 2D grid of positions, each of which records a 2D diffraction pattern by a converged electron probe. Here, we utilized abTEM (Madsen & Susi, 2021) to simulate the dataset: 10-layer-stacked molybdenum disulfide ($MoS_2$), an emergent two-dimensional semiconductor that demonstrates strong potential to exceed the fundamental limits of silicon electronics (Li et al., 2024). The $MoS_2$ dataset is simulated with intentionally exaggerated probe aberrations to pose challenges for the optimization algorithms.

**Simulation settings** The $MoS_2$ dataset uses an 80 kV probe energy, a 20 mrad probe-forming semi-angle, a set of probe aberration coefficients of defocus -130 Å, two-fold astigmatism (C12) 20 Å, two-fold astigmatism angle (Phi12) 0.785, three-fold astigmatism (C23) 15 Å, three-fold astigmatism angle (Phi23) 0.295, axial coma (C21) 30 Å, axial coma angle (Phi21) 0.534, spherical aberration (C30) $-2 \times 10^4$ Å. The dataset consists of 51 diffraction patterns with a 0.312 Å scanning step size in the real space. In addition, all diffraction patterns in both datasets were corrupted with Poisson noise of 10,000 e/Å$^2$ for this task. The ptychographic reconstruction is performed with a multi-slice approach using py4DSTEM (Savitzky et al., 2021), a comprehensive open-source package for different modes of 4D-STEM data analysis.

**Evaluation metrics** In addition to the NMSE score, we evaluate the quality of electron ptychographic reconstruction using two extra metrics: probe and object reconstruction errors. First, the probe reconstruction error calculates the normalized mean square error between the reconstructed and the simulated probes in the real space. While the ptychographic algorithm itself does not have the access to the ground truth probe function, a successful ptychographic reconstruction must accurately retrieve both the probe function and the object transmission function. As we deliberately exaggerated the aberrations of the probe in the $MoS_2$ dataset, this metric can act as another useful metric to evaluate the reconstruction. Second, the object reconstruction error computes the normalized mean

Table S1: Optimized reconstruction parameters by different AL methods for the MoS$_2$ dataset.

| | semiangle cutoff (mrad) | energy (kV) | number of iterations | step size | identical slices iteration | slice thicknesses (Å) | number of slices | defocus (Å) | C12 (Å) | phi12 (rad) | C30 (Å) | C21 (Å) | phi21 (rad) | C23 (Å) | phi23 (rad) |
|---|---|---|---|---|---|---|---|---|---|---|---|---|---|---|---|
| Ground truth | 20.0 | 80 | - | - | - | - | - | -130 | 20 | 0.79 | $-2.0\times10^4$ | 30 | 0.53 | 15 | 0.29 |
| Diff-Evo | 23.4 | 73 | 20 | 0.65 | 4 | 4.6 | 16 | -185 | 6.0 | 0.95 | $-9.4\times10^4$ | 95 | 0.06 | 84 | 1.00 |
| DA | 22.0 | 269 | 18 | 0.87 | 33 | 5.4 | 21 | -118 | 50.0 | 0.61 | $-4.1\times10^4$ | 62 | 0.15 | 47 | 0.87 |
| TuRBO5 | 18.0 | 242 | 20 | 0.57 | 2 | 18.1 | 29 | -8 | 4.0 | 0.60 | $-5.6\times10^4$ | 42 | 0.57 | 19 | 0.54 |
| BO | 22.4 | 254 | 10 | 0.71 | 2 | 34.4 | 17 | -166 | 16.0 | 0.83 | $-5.4\times10^4$ | 89 | 0.3 | 16 | 0.03 |

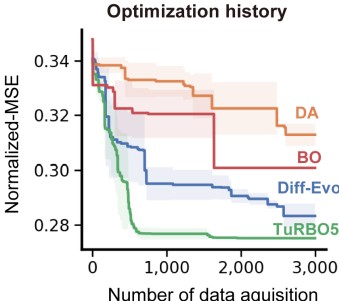

Figure S3: Optimization history of AL methods on the MoS$_2$ dataset.

square error between the median-angle-annular-dark-field signal (without added noise) and the phase of the object transmission function. This metric directly demonstrates the quality of the retrieved object transmission function.

**Hyper-parameter settings** We used 20 samples for initialization of all AL methods. We set the independent trust regions to 5 for TuRBO. The rest hyper-parameters take the default values for the individual AL methods.

**Optimization results** The optimization history (Figure S3) shows that TuRBO achieves the lowest NMSE after 500 samples, while other methods are trapped into local minima. Table S1 summarizes the reconstruction parameters for each AL method. Figure S4 visualizes the reconstructed amplitude of the probe functions with different AL methods.

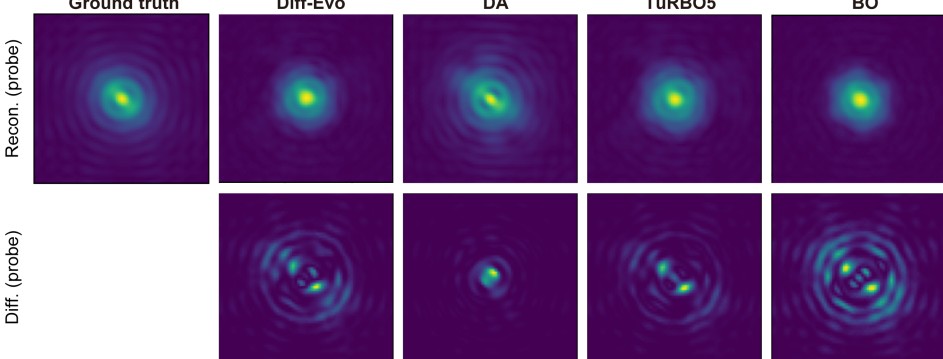

Figure S4: Visualization of amplitude (of the probe functions) reconstructed using parameters obtained from the corresponding AL methods. The second row visualizes the normalized mean square error between the ground truth and the reconstructed amplitude values.

