# OpenReview forum: "BALSA: Benchmarking Active Learning Strategies for Autonomous laboratories"
_ICLR.cc/2025/Conference — ICLR 2025 Conference Withdrawn Submission_

### Official Review · Reviewer_1ofk · 2024-11-03

**Soundness:** 3
**Presentation:** 3
**Contribution:** 1
**Rating:** 3
**Confidence:** 5

**Summary:**

This paper aims to provide benchmark problems for active learning algorithms in automatically selecting experimental conditions in a Self-Driving Laboratory (laboratory automation). Laboratory automation is a field gaining attention across various disciplines, and establishing benchmarks for its core methodology, active learning methods, is essential. In this paper, benchmark problems using six artificial functions and two simulators are examined, and experiments are conducted on eleven types of active learning methods.

**Strengths:**

1. Laboratory automation is a field gaining attention and beginning to be explored across various disciplines; however, a lack of suitable benchmark problems makes it challenging to accurately assess the effectiveness of heuristic methods proposed from different fields. Efforts to establish benchmark problems for active learning in laboratory automation are, therefore, very important to address this issue.

2. The no-free lunch theorem implies that no single method excels universally across all optimization problems; methods should be chosen based on the specific type of problem. I fully support this author's view, as well as the approach of classifying and analyzing problem types according to the landscape smoothness of optimization problems.

**Weaknesses:**

1. I do not believe that the problem settings, datasets, and methods discussed in this paper are sufficient for evaluating algorithms for laboratory automation. First, it seems necessary to verify, in some way, whether benchmark functions such as Ackley and Rosenbrock sufficiently cover the class of optimization problems in laboratory automation. Since these benchmark functions were designed to measure the effectiveness of traditional nonlinear optimization algorithms, using them directly may not be appropriate.

2. Since this paper aims to provide benchmark problems, properly evaluating its originality for Top-tier conferences such as ICLR is difficult. However, for example, the metric presented as a novel measure for landscape flatness in Equation (2) is quite naive and not particularly new. A more original perspective tailored specifically to the issues in laboratory automation would strengthen this paper.

**Questions:**

None.

---

> ### Author Response · Authors · 2024-11-21
>
> > ***Q1.** I do not believe that the problem settings, datasets, and methods discussed in this paper are sufficient for evaluating algorithms for laboratory automation. First, it seems necessary to verify, in some way, whether benchmark functions such as Ackley and Rosenbrock sufficiently cover the class of optimization problems in laboratory automation. Since these benchmark functions were designed to measure the effectiveness of traditional nonlinear optimization algorithms, using them directly may not be appropriate.*
>
> Thank you for your comments regarding the suitability of traditional benchmark functions for evaluating algorithms in laboratory automation. We understand your concern about whether these functions sufficiently cover the optimization problems encountered in the self-driving lab tasks. While these functions were originally designed for traditional optimization algorithms, we have adapted them to reflect the distinct constraints of real-world problems by **incorporating limited data availability and the use of surrogate models within an active learning framework**. This adaptation enables us to systematically evaluate algorithms under conditions that are representative of real-world applications, such as the necessity of searching with sparse data.
>
> Our purpose in using these synthetic functions is to **provide a unified framework with easy-to-measure sample efficiency and known optima**. These functions allow us to simulate different topologies and assess various active learning pipelines in a low-data regime - a setting that resembles the constraints of self-driving lab tasks. It is important to note that these functions have been widely used in the black-box optimization community [1,2], and our approach builds on this foundation to address the specific challenges of real-world tasks.
>
> Additionally, **we have included two real-world tasks to simulate real-world scenarios**. While we recognize that these tasks may not cover the entire spectrum of self-driving lab challenges, they are accessible and open-source, allowing for reproducibility and wider community engagement. Due to constraints with closed-source software and data, we are limited in the range of tasks we can include, but we have tried our best to select the most relevant and practical ones.
>
> We acknowledge that synthetic functions cannot capture all the complexities of real-world self-driving lab problems. In future work, we plan to expand our benchmark suite to include more diverse and representative problem settings.
>
> We appreciate your feedback and hope that our clarifications address your concerns regarding the applicability of these benchmark functions.
>
>
> *[1] D. Brockhoff, T. Tusar, A. Auger, N. Hansen, Using well-understood single-objective functions in multiobjective black-box optimization test suites.*
>
> *[2] D. Eriksson, M. Pearce, J. R. Gardner, R. Turner, M. Poloczek, Scalable global optimization via local bayesian optimization (2019).*

---

> > ### Author Response · Authors · 2024-11-21
> >
> > > ***Q2.** Since this paper aims to provide benchmark problems, properly evaluating its originality for Top-tier conferences such as ICLR is difficult. However, for example, the metric presented as a novel measure for landscape flatness in Equation (2) is quite naive and not particularly new. A more original perspective tailored specifically to the issues in laboratory automation would strengthen this paper.*
> >
> > Thank you for your valuable feedback regarding the landscape flatness metric and the originality of our work. We acknowledge the landscape flatness metric presented in *Equation (2)* is a simple empirical measure and might not be novel. Its purpose is to provide additional insight into the optimization landscapes we are examining, rather than to serve as a primary contribution.
> >
> > The originality of our work lies in proposing a benchmark suite and two open-source real-world problems that address the specific challenges of self-driving lab tasks. Specifically, we focus on **limited data availability** and **the use of surrogate models** within an **active learning pipeline** - a novel perspective tailored to self-driving lab tasks. Self-driving lab tasks are different from standard optimization scenarios because they can be formulated as iterative optimization under severe data constraints. In these real-world tasks, the landscape of the complex system is often unknown, and obtaining new data is both costly and time-consuming. **Standard optimization algorithms may struggle under these constraints** when searching for optimal designs. To overcome these challenges, we employ an iterative active learning strategy where algorithms search for optimal designs within surrogate-model-fitted landscapes before validation with oracle functions. This approach reflects the practical workflow in self-driving lab tasks, where surrogate models guide experimentation under tight data constraints.
> >
> > Our benchmarks on synthetic functions for the active learning pipeline have been deliberately designed to capture the following criteria:
> >
> > * **Iterative process**: mimicking the step-by-step approach used in real-world self-driving lab settings.
> > * **Use of surrogate models**: leveraging surrogate models to approximate complex systems in scenarios with scarce data.
> > * **Low-data regime**: emphasizing the challenges of limited data availability inherent in real-world tasks.
> >
> > By considering these specific aspects, our work provides a benchmarking framework that encapsulates the unique difficulties of self-driving lab tasks. In this case, search algorithms that perform well on our synthetic benchmarks could potentially offer valuable insights into real-world applications.

---

> > > ### Comment · Reviewer_1ofk · 2024-12-03
> > >
> > > I appreciate the author's reply to my comments. I fully agree with the authors' perspective that benchmarking for laboratory automation is important, and I believe this paper is valuable as a first step. However, as a paper for a top conference like ICLR, there seems to be considerable room for improvement. Therefore, I will keep the score as it is.

---

> > ### Comment · Reviewer_1ofk · 2024-12-03
> >
> > I appreciate the author's reply to my comments. However, I cannot agree with the claim that evaluations on traditional benchmark functions such as Ackley and Rosenbrock are sufficient for benchmarking in laboratory automation. Therefore, I keep the score as it is.

---

### Official Review · Reviewer_MEVR · 2024-11-03

**Soundness:** 2
**Presentation:** 2
**Contribution:** 2
**Rating:** 5
**Confidence:** 3

**Summary:**

BALSA is a paper introducing new benchmarks for active learning in autonomous labs, both in the context of protein design and materials design. However, I get the feeling that the authors have tried to do too much, and I found the paper to be hard to read and poorly organized, though the ideas are certainly interesting. I think the paper could benefit from significant restructuring, and possibly even first perfecting the benchmark for protein design only, then extending to materials (or vice-versa); in its current state, it feels imperfect for both, and I did not feel that the paper adequately reviewed either existing protein design benchmarks or existing materials design benchmarks, making it hard to understand what is the value added here by BALSA. Datasets and benchmarks is perhaps the hardest track of papers in which to get accepted, as authors have to make things simple yet rigorous (i.e., foolproof) and transparent for other researchers to use, and I do not think BALSA in its current state meets this criteria.

**Strengths:**

* It is a good ambition to try and develop a suite of benchmarks that aim to ensure reproducibility of algorithmic performance across a wide variety of synthetic and real-world tasks.
* I like the idea of having the six functions as part of the benchmark (Ackley, Rastrigin, Rosenbrock, Griewank, Schwefel, and Michalewicz), but overall I think the benchmark in its current state does not seem general or rigorous enough to qualify for acceptance at ICLR.
* The plots themselves are very nice.

**Weaknesses:**

* Unfortunately, the anonimized repository does not link to anything, which is a shame because it means the benchmarks introduced here are not reproducible.
* Not sure if AF2 is the most relevant model for evaluating the protein design tasks, as we are generally trying to push models outside distribution when designing new proteins, which is precisely the scenarios in which AF2 would fail. Furthermore, the materials benchmark proposed herein would only apply to crystalline materials, limiting its applicability.
* It is not fully clear to me what is the novelty of the benchmarks introduced here, relative to previous benchmarks.
* The paper could benefit from a more thorough proofreading, including of the formatting which was strange in places throughout the text and distracting. For instance, the way references were inserted in the text made little sense to me and it as hard to understnad what part of the sentence was being referenced.

**Questions:**

* In active learning for molecular optimization (regardless of protein or materials), one of the most crucial components we seek to optimize is sample efficiency; however, it was not clear to me how this is assessed in this paper, even though the authors say this is what they are trying to assess. For instance, none of the presented results touch on sample efficiency (and, if they do, this was not clear). Can this be clarified?
* The figure and table captions are not informative, and could be improved for clarity. For instance, what are the values that are being shown in Table 1, is higher/lower better, and what are the bounds? Same for Table 2. What the values are could be made clear in the captions, without needing to go dig through the text (and, I could not always find what it was that was being shown in the tables).

---

> ### Author Response · Authors · 2024-11-20
>
> > ***Q1.** Unfortunately, the anonimized repository does not link to anything, which is a shame because it means the benchmarks introduced here are not reproducible.*
>
> Thank you for bringing this to our attention. We are sorry for the inconvenience caused by the inaccessible anonymized repository. Due to the double-blind review process, the repository link was anonymized, which unfortunately made it inaccessible. **We have now included the complete repository as an attachment with this revision (in the supplementary material)** to ensure full reproducibility of our benchmarks.
>
> > ***Q2.** Not sure if AF2 is the most relevant model for evaluating the protein design tasks, as we are generally trying to push models outside distribution when designing new proteins, which is precisely the scenarios in which AF2 would fail. Furthermore, the materials benchmark proposed herein would only apply to crystalline materials, limiting its applicability.*
>
> We appreciate the reviewer’s insightful comments. Regarding the biology task, our focus is more specific than general protein design - **it involves designing a specialized type of protein with therapeutic applications**. This protein is required to exhibit stronger interactions with its target, such as higher binding affinity. This task can be framed as an optimization problem. However, even for a relatively simple 16-residue sequence, the combinatorial search space includes $16^{20}$ possible configurations. The intricate and nonlinear relationship between protein sequence and functional properties further complicates the challenge, making it a suitable benchmark for testing advanced methodologies. An additional advantage of this setup is the availability of natural binders as a reference for comparison.
>
> Regarding the materials benchmark, we understand the concern about its applicability. **Electron ptychography has been successfully applied to a wide range of materials** [1], including amorphous [2] and organic–inorganic hybrid nanostructures [3]. This broad applicability ensures that our benchmark is relevant to various material types, not limited to crystalline structures.
>
> [1] P. Wang, F. Zhang, S. Gao. et al. Electron Ptychographic Diffractive Imaging of Boron Atoms in LaB6 Crystals. Scientific Report 7, 2857 (2017).
>
> [2] S. Karapetyan, S. Zeltmann, T.-K. Chen, et al. Visualizing Defects and Amorphous Materials in 3D with Mixed-State Multislice Electron Ptychography, Microscopy and Microanalysis (2024).
>
> [3] Z. Ding, S. Gao, W. Fang, et al. Three-dimensional electron ptychography of organic–inorganic hybrid nanostructures. Nature Communication 13, 4787 (2022).
>
> > ***Q3.** It is not fully clear to me what is the novelty of the benchmarks introduced here, relative to previous benchmarks.*
>
> We appreciate the opportunity to clarify the novelty of our work. A key contribution of our work is that **we introduce a benchmark that explicitly incorporates the limitation of data acquisition** into the active learning pipeline - i.e., the pipeline has only limited access to ground truth labels. This constraint simulates a common challenge in scientific problems, where data labeling is expensive and time-consuming in real-world scenarios.
>
> In addition, due to the availability of the open-source software, we focused on tasks that are both challenging and accessible for the broader community. While more complex tasks like designing compositionally complex alloys (CCAs) would be well-suited as a self-driving lab task - where the alloy properties are optimized by adjusting the alloy compositions - the simulations to assess these properties require licensed software. This limitation makes such tasks less suitable for open-source benchmarking purposes.
>
> > ***Q4.** The paper could benefit from a more thorough proofreading, including of the formatting which was strange in places throughout the text and distracting. For instance, the way references were inserted in the text made little sense to me and it as hard to understnad what part of the sentence was being referenced.*
>
> We appreciate and agree with this comment and are in the process of reformatting the paper, particularly the references.

---

> > ### Author Response · Authors · 2024-11-20
> >
> > > ***Q5.** In active learning for molecular optimization (regardless of protein or materials), one of the most crucial components we seek to optimize is sample efficiency; however, it was not clear to me how this is assessed in this paper, even though the authors say this is what they are trying to assess. For instance, none of the presented results touch on sample efficiency (and, if they do, this was not clear). Can this be clarified?*
> >
> > We thank the reviewer for pointing this out. To clarify how we assess sample efficiency, we plan to include history plots (or progress plots) for the selected examples in the main text as Figure 7 (may put part of the plots in the supplementary material, depending on the page limit) for further clarification. We are currently preparing these modifications and will incorporate them in the revised manuscript.
> >
> > > ***Q6.** The figure and table captions are not informative, and could be improved for clarity. For instance, what are the values that are being shown in Table 1, is higher/lower better, and what are the bounds? Same for Table 2. What the values are could be made clear in the captions, without needing to go dig through the text (and, I could not always find what it was that was being shown in the tables).*
> >
> > We thank the reviewer for this excellent comment. We entirely agree with this and are in the process of reformatting the corresponding tables and captions to improve the clarity.

---

> ### Comment · Reviewer_MEVR · 2024-11-25
>
> Thank you to the reviewers for their thoughtful improvement of the paper and study based on the suggested changes. Thank you also for providing the code. The submission is stronger now.
>
> I have updated my score accordingly!
>
> I also would recommend to add more documentationto the code if possible, perhaps some tutorials, to increase the utility to potential users.

---

> > ### Author Response · Authors · 2024-11-26
> >
> > Thank you for your prompt feedback and for recognising the improvements in the submission.
> >
> > We completely agree that adding more documentation enhances the usability of the code. To address this, we have:
> > * **Enhanced `README.md`** and included a new tutorial section available in `./docs/tutorials`.
> > * **Added a tutorial `notebook`** that demonstrates key functionalities, making it easier for new users to get started.
> >
> > The updated code and documentation are included in the supplementary materials.
> >
> > We are glad that our revisions addressed your concerns, and we truly appreciate your encouraging comments. Please do let us know if you have any additional suggestions!

---

### Official Review · Reviewer_ycRe · 2024-11-09

**Soundness:** 2
**Presentation:** 2
**Contribution:** 3
**Rating:** 6
**Confidence:** 3

**Summary:**

This paper introduces a benchmark suite for evaluating active learning strategies in autonomous laboratories. The work includes synthetic benchmark functions, two real-world tasks (protein design and electron microscopy), and introduces a new metric called landscape flatness to characterize objective functions. The authors evaluate 11 baseline methods on synthetic tasks and 4 methods on real-world applications.

**Strengths:**

* The authors provide a broad set of tasks, from synthetic to complex real-world scenarios in biology and materials science. The authors explain clear difference from traditional optimization benchmarks.
* The authors introduce the landscape flatness metric, which can quantify the complexity of the objective landscape.
* The authors provide detailed experiments with 11 baseline methods.

**Weaknesses:**

* The authors do not provide enough validation for the proposed landscape flatness metric. It would be better to have theoretical evidence to support the robustness of this metric across different tasks.
* For experiments, the authors do not explain different numbers of trials (5 trials for synthetic tasks, 3 trials for real-world tasks). Results in Table 1 show high variance across trials, but the authors do not discuss this variation.
* The authors highlight the scalability as a key contribution, but the paper's analysis of this aspect is limited. For example, there is no quantitative analysis of how computation time scales with dimensionality. And the maximum dimension is 100D.
* Others:
    * It is hard to understand the figures (e.g., Fig. 2, 6) due to small size, unclear labeling and limited context.
    * The introduction contains redundant information about self-driving labs; Minor typo errors and inconsistencies are present in the paper.

**Questions:**

See Weaknesses.

---

> ### Author Response · Authors · 2024-11-20
>
> > ***Q1.** The authors do not provide enough validation for the proposed landscape flatness metric. It would be better to have theoretical evidence to support the robustness of this metric across different tasks.*
>
> Thank you for bringing this important point to our attention. We appreciate your suggestion regarding the landscape flatness metric. While our primary contributions are the benchmark suite and the two real-world problems, **we agree that providing theoretical evidence would enhance the robustness of this metric**. However, a full theoretical analysis is beyond the scope of this paper. The landscape flatness is an empirical observation, and measuring it helps us gain insights into these high-dimensional distributions.  We acknowledge your concern and plan to move the relevant discussion of landscape flatness to the supplementary material for clarity. We are currently revising the manuscript to reflect this change. Strengthening the theoretical work regarding the flatness metric could be a promising future direction.
>
> > ***Q2.** For experiments, the authors do not explain different numbers of trials (5 trials for synthetic tasks, 3 trials for real-world tasks). Results in Table 1 show high variance across trials, but the authors do not discuss this variation.*
>
> Thank you for pointing out the lack of explanation regarding the different numbers of trials and the observed variance. We used 5 trials for synthetic tasks due to their lower computational cost, allowing for more extensive experimentation. For the real-world tasks, computational constraints limited us to 3 trials. We are adding this explanation into the discussion section and will include it in the revised manuscript.
>
> **The observed variance is mainly due to data sparsity caused by high dimensionality.** In our active learning pipeline, we train a surrogate model which will then be explored by search algorithms for optimization. Since the search algorithm doesn’t have direct access to the ground truth labels, the random initialization of the training dataset for the surrogate model influences the final results. Different random initializations lead to different surrogate models, resulting in higher variance across trials. Higher-dimensional problems are expected to exhibit higher variance. We have added this into the discussion section.
>
> > ***Q3.** The authors highlight the scalability as a key contribution, but the paper's analysis of this aspect is limited. For example, there is no quantitative analysis of how computation time scales with dimensionality. And the maximum dimension is 100D.*
>
> Thank you for highlighting the need to clarify our use of the term ‘*scalability.*’ We agree that the phrase ‘*scalability*’ requires further clarification. Our intention was to convey that our active learning pipeline is applicable not only to problems with tens of dimensions, but also to those with hundreds of dimensions.
>
> As you correctly pointed out, our key contribution lies in proposing methods to handle problems with limited data availability. The key challenge in self-driving lab tasks is **data acquisition** rather than **computation time scaling**. In these problems, the available data is often just a few hundred samples, and the limited data sampling rate is the real bottleneck instead of the computational time. This is because conducting experiments or simulations can be extremely costly; processes such as synthesizing and characterizing advanced alloys or drug-relevant molecules can cost millions of dollars and take months or even years of intense labor. The essential goal of AI-driven self-driving labs is to iteratively identify and label the most informative data points to discover the next best candidates while **minimizing data labeling efforts**.
>
> We are revising the key contribution in the main text to address your feedback.
>
> > ***Q4.** Others:
> It is hard to understand the figures (e.g., Fig. 2, 6) due to small size, unclear labeling and limited context.
> The introduction contains redundant information about self-driving labs; Minor typo errors and inconsistencies are present in the paper.*
>
> Thank you for pointing out these areas where we can further improve. We are addressing these concerns in our revisions and will incorporate the changes in the updated manuscript.

---

### Author Response · Authors · 2024-11-20

We would like to thank all reviewers for your thoughtful feedback on our paper introducing BALSA. We are pleased that you found our work to present broad tasks spanning synthetic and real-world scenarios with clear differentiation from traditional benchmarks (Reviewer `ycRe` and `MEVR`). Furthermore, we are particularly encouraged by your recognition of the new evaluation metrics that better characterize the objective landscape, and our comprehensive empirical study, which covers both synthetic and real-world design tasks (Reviewer `ycRe`, `MEVR` and `1ofk`).

We appreciate your feedback regarding the novelty and relevance of the synthetic functions to laboratory automation. Benchmarking AI-driven scientific problems is particularly challenging due to the limited data availability (inherently expensive evaluations). Our work addresses this by tailoring an active learning pipeline to reflect real-world constraints, such as **iterative optimization** under **severe data limitations** and **the use of surrogate models**. These additions differentiate our synthetic benchmarks from traditional optimization functions and ensure their relevance to laboratory automation. We hope this could further clarify the novelty of our benchmark.

We acknowledge the concerns about the insufficient theoretical validation for the landscape flatness metric. While a full theoretical analysis of the landscape flatness metric is beyond the scope of this paper, we agree that such evidence would enhance the impact of this metric. We aim to investigate this further in future work and will include preliminary insights in the supplementary material. We plan to move part of the content into the supplementary material while retaining essential information about this metric, as we believe the flatness metric offers valuable insights into the landscape complexity.

To address the reproducibility concern, we sincerely invite you to check out our open-source implementations and technical details. We have attached the complete repository along with this submission.

In response to your comments, we are carefully addressing each point and are in the process of revising our manuscript accordingly. **We would like to provide this initial responses first and will upload the revised version incorporating all suggested changes soon.** We hope that our revisions address all your concerns and meet the expectations of the conference committee.

Once again, we sincerely appreciate the time and effort you have taken to review our paper. We agree that the communication of the work could have been better, and are more than willing to add further clarifications to address any additional recommendations or feedback from you.

---

> ### Author Response · Authors · 2024-11-28
> **Summary of revisions and key improvements in the manuscript**
>
> We appreciate the insightful feedback provided during the review process. We have **uploaded the revised manuscript** incorporating the feedback provided. The revised version directly addresses the feedback and key concerns raised by the reviewers. Here we summarize some key improvements and changes:
>
> - Clarified the **unique contributions** of the proposed active learning pipeline tailored for self-driving labs, which includes:
>     - `Iterative nature` that mirrors real-world processes by guiding experimentation step-by-step
>     -  `Integration with surrogate models`, enabling efficient approximations of complex systems.
>     - `Incorporation of limited data availability` that simulates the real-world expensive data acquisition
>
> - Expanded the evaluation scope by **introducing two additional real-world benchmark tasks**:
>     - `Neural Network Architecture Search (NAS)`: Reformulated as a benchmark for performance optimization in AL pipelines
>     - `Lunar Landing Problem`: Adapted as a discrete optimization task to evaluate AL pipelines in control dynamics
>
> - Other key improvements include:
>     - Detailed analyses of sample efficiency, providing a deeper evaluation of algorithm performance on synthetic functions and real-world tasks
>     - Enhanced clarity of captions and tables, improving readability and comprehension
>     - Relocation of technical details to supplementary materials, streamlining the main text and increasing accessibility of key findings
>     - Providing the source code ensuring reproducibility

---

### Note · Authors · 2025-01-23

I have read and agree with the venue's withdrawal policy on behalf of myself and my co-authors.